# Improvement in Quality of Life through Self-Management of Mild Symptoms during the COVID-19 Pandemic: A Prospective Cohort Study

**DOI:** 10.3390/ijerph19116652

**Published:** 2022-05-30

**Authors:** Ryuichi Ohta, Yoshinori Ryu, Chiaki Sano

**Affiliations:** 1Community Care, Unnan City Hospital, 96-1 Iida, Daito-cho, Unnan 699-1221, Japan; yoshiyoshiryuryu.hpydys@gmail.com; 2Department of Community Medicine Management, Faculty of Medicine, Shimane University, 89-1 Enya-cho, Izumo 693-8501, Japan; sanochi@med.shimane-u.ac.jp

**Keywords:** help-seeking behavior, self-care, quality of life, rural community, EQ-5D-5L, COVID-19

## Abstract

The COVID-19 pandemic has inhibited people’s help-seeking behaviors (HSBs). In particular, older people in rural communities experienced limited access to medical care, which negatively affected their quality of life (QOL). Within HSB, self-management of mild symptoms may mitigate the difficulties experienced by older people in rural communities. However, few studies have examined the relationship between self-management and QOL. Therefore, we conducted a prospective cohort study to clarify this relationship. Our participants were over 65 years of age and lived in rural communities. QOL was measured with the EuroQol 5-Dimension 5-Level (EQ-5D-5L). Demographic data and QOL were collected from participants via questionnaires in 2021 and 2022. The exposure group showed a significantly greater change in EQ-5D-5L health status index scores than the control group (*p* = 0.002). In addition, the exposure group scored significantly lower than the control group on the EQ-5D-5L dimension “usual activities” in 2021 and on all dimensions in 2022. Thus, self-management of mild symptoms may improve QOL among older people in rural communities during the COVID-19 pandemic. Educational interventions for this population regarding self-management could improve QOL for entire communities.

## 1. Introduction

The COVID-19 pandemic has inhibited people’s help-seeking behaviors (HSBs). Specifically, many people were unable to access healthcare resources effectively due to travel restrictions and, therefore, had to endure or manage symptoms by themselves. HSB is a human behavior aimed at maintaining health and seeking treatment for symptoms [1]. HSB should be utilized effectively, as it is important for managing health conditions [2].

A balance of lay and professional care is important for patients to deal with mild and severe symptoms [3]. Mild symptoms refer to symptoms that people feel are manageable without healthcare support. Severe symptoms need healthcare support immediately. Lay care refers to unpaid healthcare provided by laypeople who have received no formal training, such as self-care and care from relatives, friends, and self-help groups [4]. In contrast, professional care refers to healthcare provided by trained, paid professionals, usually in a formal setting [4]. Efficient utilization of lay and professional care can reduce the overuse of professional care for mild symptoms [5,6]. Effective HSB for mild symptoms can moderate disease progression through early detection [5,6].

More than 70% of older adults have chronic diseases with high potential for severe complications [7]. During the pandemic, older people were forced to manage their health conditions differently than usual due to limited access to medical care. This might have caused anxiety and depression in these patients, which, in turn, may have affected their health conditions.

Older people in rural areas experienced particularly limited access to medical care; as COVID-19 posed a high risk to this population, this negatively affected their quality of life (QOL). QOL is defined as a person’s perception of the conditions in their lives in the context of their culture and value systems. Older people suffer from various chronic diseases that require appropriate treatment from established healthcare systems [8]. As the number of visits to medical institutions increases, medical expenses increase as well and create financial challenges [4]. To prevent disease progression, primordial and tertiary prevention in primary care are essential to effectively manage health conditions among older people [9].

HSB is influenced by various factors, including biological, social, economic, and cultural aspects [10,11,12,13]. Older people in rural areas tend to care for themselves or visit medical institutions without consulting others [14]. Previous research indicates that more than half of the people in rural communities avail themselves of professional healthcare services every month [15]. Moreover, Japanese culture impacts HSB because Japanese health insurance supports a range of medical tests and procedures; therefore, many older individuals routinely visit medical institutions, even when they only have mild symptoms [16].

The COVID-19 pandemic inhibited HSB, especially among older people in rural areas, as this population tends to use healthcare institutions within, rather than outside of, their communities [17]. QOL may be associated with HSBs, in particular, self-management and professional care usage [18,19,20]. A previous study conducted in rural areas showed that self-management of mild symptoms was associated with better self-rated health [21]. However, there is no clear evidence regarding the relationship between self-management of symptoms and QOL.

Within HSB, self-management of symptoms may mitigate the difficulties experienced by older people in rural communities. To ensure a sustainable healthcare system during the COVID-19 pandemic, self-management of mild symptoms among older people in rural areas is essential [22]. In rural communities, healthcare resources are declining, and healthcare professionals are aging [23,24], which inhibits access to professional care. The COVID-19 pandemic has the potential to overwhelm rural healthcare; therefore, self-management of mild symptoms is important. As many people live in rural areas worldwide, an investigation regarding HSB in such communities during the COVID-19 pandemic is beneficial for global healthcare [25].

Clarifying the relationship between self-management and QOL may facilitate more effective use of healthcare resources. Furthermore, educational materials could be developed and provided to rural communities systemically through top-down and upstream interventions. Per month, more than 70% of the rural population experiences mild health symptoms that can be managed through lay care [26,27]. Thus, effective self-management may mitigate the burden on professional care during the COVID-19 pandemic. However, few studies have investigated the relationship between self-management and QOL. Therefore, this study examined the relationship between self-management of mild symptoms and QOL among older people in rural communities during the COVID-19 pandemic.

## 2. Materials and Methods

### 2.1. Setting

This prospective cohort study was conducted in Kakeya and Yoshida in Unnan, Japan. A survey conducted in 2017 revealed that the total population of Unnan was 38,882 (18,720 males, 20,162 females) and that the aging rate was 37.82%. This rate is estimated to reach 50% by 2050. Kakeya is a town located in the westernmost region of Unnan and consists of five districts: Kakeya, Matsukasa, Tane, Iruma/Anami, and Hata. Yoshida is a town located in the southeastern region of Unnan and consists of two districts: Tai and Yoshida. Participants from the Kakeya, Matsukasa, Tane, and Tai districts were included in the study.

### 2.2. Participants

The participants were over 65 years of age and lived in Kakeya, Matsukasa, Tane, or Tai. The total populations of people over 65 years of age in Kakeya, Matsukasa, Tane, and Tai were 554, 230, 109, and 173, respectively. The participants were contacted by a written letter that included an explanation of the study and a research questionnaire. Individuals who were unable to read or write as well as those with dementia were excluded from the study. To measure changes in QOL, the questionnaire was completed in both February 2021 and February 2022 by all participants. As our study focused on the association between QOL and self-management of mild symptoms, respondents who showed a preference to self-manage their symptoms in 2021 were excluded. The included participants were divided into two groups: (1) participants who acquired a preference for self-management of mild symptoms (exposure group) and (2) participants who did not acquire a preference for self-management of mild symptoms (control group).

### 2.3. Measurements

#### 2.3.1. Outcome Variable

QOL was our primary outcome variable and was measured using the EuroQol 5-Dimension 5-Level (EQ-5D-5L). The EQ-5D-5L is a self-administered questionnaire consisting of five dimensions: (1) mobility, (2) self-care, (3) usual activities, (4) pain/discomfort, and (5) anxiety/depression. The question regarding mobility asks about problems in walking. Regarding self-care, the question asks about problems in walking or dressing oneself. Regarding usual activities, the question asks about problems in performing one’s usual activities such as work, study, housework, familial duties, or leisure activities. Regarding pain/discomfort, the question asks about having pain or discomfort in one’s usual life. Finally, the question regarding anxiety/depression asks about any feelings of anxiety/depression.

Each dimension was measured according to five levels of severity, namely (1) no problems, (2) slight problems, (3) moderate problems, (4) severe problems, and (5) extreme problems.

These scores were subsequently combined into five-digit numbers representing the participants’ health status profiles. Health status profiles were converted into single health status index scores through the application of a formula that attached values to each response. The Japanese version of the EQ-5D-5L has been validated (R^2^ = 0.977) [28].

#### 2.3.2. Independent Variable

A validated questionnaire with Spearman’s ρ = 0.707 and Cohen’s kappa = 0.836 was used to measure trends in participants’ HSBs for mild symptoms [29]. In a previous study, we established the use of the questionnaire to inquire about HSB for mild symptoms among older people and checked its validity and reliability. In this questionnaire, participants reported their behavioral responses to mild symptoms. The previous study used data from older people in various settings, such as clinics and hospitals. In this study, we focused on rural communities and defined the presence of self-management (changing lifestyles, sleeping, resting, and taking a bath) in the questionnaire as showing a preference to self-manage mild symptoms.

#### 2.3.3. Covariates

Participants’ background information, including age, sex, body mass index (BMI), smoking, alcohol use, education level, living conditions, social support, and socioeconomic status (SES), was collected via the questionnaire [30].

### 2.4. Statistical Analyses

Parametric data were analyzed using the Student’s *t*-test, and categorical data were analyzed using the chi-squared test. The independent variables were categorized binomially: sex (male = 1, female = 0), smoking (yes = 1, no = 0), alcohol use (yes = 1, no = 0), education level (high school or above = 1, less than high school = 0), living conditions (with family = 1, alone = 0), self-rated health (good or relatively good = 1, relatively bad or bad = 0), social support (present or relatively present = 1, relatively not present or not present = 0), and SES (high = 1, low = 0). A significance level of *p* < 0.05 was used for all comparisons. A minimum of 64 participants were required in each group based on α (alpha) = 0.05, β (beta) = 0.20 (power of 80%) to measure the statistical difference of change in EQ-5D-5L single health status index scores as the mean and standard deviations of the difference of change between groups were 0.08 and 0.16, respectively. [21,31]. The covariate balance between the matched groups was examined. All statistical analyses were performed using EZR (Saitama Medical Center, Jichi Medical University, Saitama, Japan), which is a graphical user interface for R (The R Foundation, Vienna, Austria) [32].

### 2.5. Ethical Considerations

Participants were informed that the data collected in the study would only be used for research purposes. They were also informed of the aims of the study as well as the ways in which data would be disclosed and their personal information protected. Participants then provided written consent. The study was conducted in accordance with the principles of the Declaration of Helsinki and was approved by the Unnan City Hospital Clinical Ethics Committee (approval number: 20200013).

## 3. Results

### 3.1. Demographic Data of the Participants

In 2021, the total effective response rate of the questionnaires was 78.2% (834/1066). Of the total respondents, 631 were excluded due to missing information in 2022 (*n* = 310) or a preference for self-management in 2021 (*n* = 321) (Figure 1). There were no statistical differences in demographic data between the exposure and control groups (Table 1).

### 3.2. Change in the Single Health Status Index Scores on the EQ-5D-5L

The change in the single health status index scores on the EQ-5D-5L was significantly higher for the exposure group than for the control group (*p* = 0.002). In addition, the exposure group scored significantly lower than the control group on the EQ-5D-5L dimension “usual activities” in 2021 and on all dimensions in 2022 (Table 2).

## 4. Discussion

This prospective cohort study showed that self-management of mild symptoms improved QOL among older people in rural areas during the COVID-19 pandemic. Our results indicate that those in this population who are motivated to manage their symptoms independently can improve their QOL. In rural areas without sufficient healthcare resources, community-wide interventions focused on the development of self-management skills in older people should be implemented.

Self-managing symptoms can improve QOL among older people in rural areas during the COVID-19 pandemic. Our findings showed that a preference for self-management was correlated with improved QOL among older people in rural communities. During the COVID-19 pandemic, this population was required to manage their own health conditions due to limited access to relatives, friends, and medical institutions [33,34,35]. A previous study showed that life difficulties could motivate people to be proactive [36,37]. In addition, people motivated to control their health conditions were able to effectively manage their health through the development of self-efficacy [38,39].

Our results indicated that older people in rural areas with high QOL were able to motivate themselves to self-manage mild symptoms during the COVID-19 pandemic. We also showed that participants who developed a preference for self-management reported better QOL in 2021 than those who did not, particularly in the dimensions “usual activities” and “pain/discomfort”. The dimension “usual activities” measures the ability of people to complete daily life tasks, such as work, study, housework, and leisure, without difficulty [28,40]. The dimension “pain/discomfort” measures the degree to which people feel pain and/or discomfort in their everyday lives [28,40]. High scores on these dimensions might affect HSBs, as people who are unable to perform everyday activities without pain and/or discomfort may not be motivated to act independently.

Amid the COVID-19 pandemic, it is essential to investigate how to motivate older people in rural areas to self-manage their health conditions, as this is vital to their QOL. Our results showed that motivated individuals in this population were able to improve their QOL despite the COVID-19 pandemic. Research has shown that older people in rural communities were able to improve their motivation to act independently, which contributed positively to their QOL. The pandemic has infringed on the lives of older people in rural areas through the implementation of infection control measures based on urban standards [17,41]. To increase self-management within this population, it may be necessary to provide support for people with low QOL [42]. Information regarding the assessment of and approaches to mild symptoms can be vital for their self-management of them. Health literacy and isolation could have affected the behaviors of those who were not motivated to engage in self-management of mild symptoms in this study. Rural societies have been suffering from isolation due to the COVID-19 pandemic, causing poor health outcomes [43]. Additionally, rurally isolated people with frailty may face difficulties while approaching information resources due to their chronic disease(s) [43]. Rural social resources regarding knowledge of health management and HSBs should be continuously informed and used to facilitate the efficient self-management of symptoms. Health care professionals should support self-management in these individuals with the collaboration of volunteers in communities who motivate people to manage health with effective exercise and nutrition [44,45].

Improvements in HSBs require social resources to continuously develop educational materials and programs. In rural areas, there are many social resources that can be used for health promotion, as there are strong social relationships among people in these regions [33,46,47]. To improve understanding of the implications of self-efficacy and/or intentions for HSBs, continuous dialogues regarding HSBs could be helpful [12]. Voluntary community meetings can be used to facilitate such dialogues [48,49]. Thus, rural areas may establish concrete activities aimed at improving HSBs among community members [50,51]. Furthermore, to empower rural community members and healthcare professionals, these people should be involved in such conversations [50,51]. Through these dialogues, older people, community workers, and healthcare professionals in rural areas may clarify their understanding of HSBs and healthcare resources [50,51,52]. In addition, they might be empowered and motivated to learn HSBs from healthcare professionals. Such interactions may contribute to the establishment of contextually appropriate educational systems for HSBs.

## 5. Conclusions

Self-management of mild symptoms can improve QOL among older people in rural areas during the COVID-19 pandemic. Educational interventions for this population regarding self-management could improve QOL for entire communities.

The limitations of this study include its prospective cohort design, which may not clearly indicate the cause-and-effect relationship between self-management and QOL. In addition, the study was conducted during the COVID-19 pandemic, and QOL may have been lower than in the previous studies [52,53,54,55]. It may not have been difficult to improve QOL during this period. Thus, the reason for the improvement in QOL in this research could be possibly explained by acquiring the preference for self-management. Because our study was conducted in super-aged rural communities in Japan, selection bias may have occurred. The demographic data showed no difference between the exposure and control groups; however, variables such as participants’ medical conditions and objective health data were not measured. Future studies should use larger samples with a wider range of demographic data to address these limitations.

## Figures and Tables

**Figure 1 ijerph-19-06652-f001:**
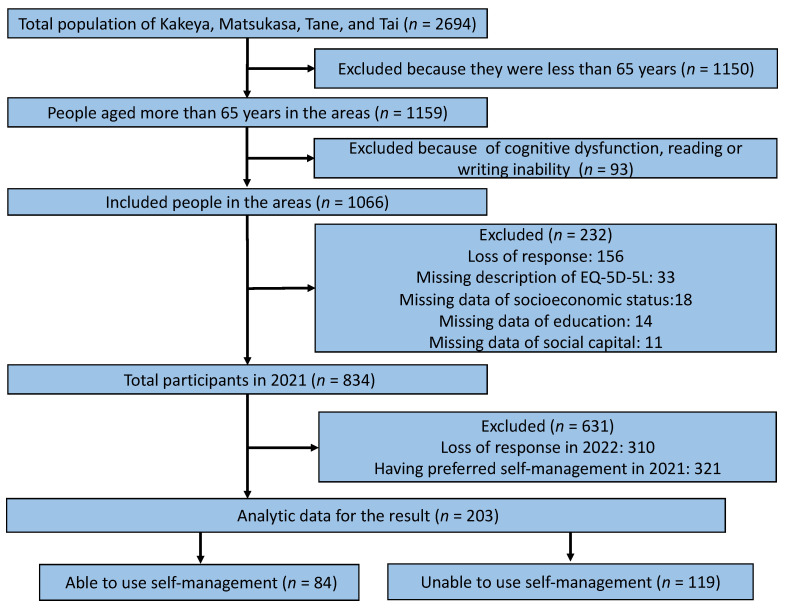
Flowchart of participant selection.

**Table 1 ijerph-19-06652-t001:** Demographic data of participants in the exposure and control groups and the significance level of the comparison between the two groups.

Variables	Using Self-Management (*n* = 84)	Not Using Self-Management *(n* = 119)	*p*-Value
Age, mean (SD)	77.26 (8.30)	78.50 (7.17)	0.26
Sex, male (%)	38 (45.8)	60 (51.3)	0.47
Weight (kg), mean (SD)	55.39 (11.75)	56.19 (11.11)	0.629
Height (cm), mean (SD)	156.39 (9.75)	156.66 (9.26)	0.844
Chronic Diseases (%)	74 (92.5)	107 (91.5)	1
Alcohol Use (%)	27 (32.1)	43 (36.4)	0.552
Smoking (%)	8 (9.5)	10 (8.4)	0.806
Higher Education (%)	35 (42.2)	51 (43.2)	1
Living with Family (%)	74 (90.2)	106 (90.6)	1
Annual Health Check (%)	63 (76.8)	88 (73.9)	0.74
High Socioeconomic Status (%)	45 (54.9)	57 (48.3)	0.39
High Social Support (%)	71 (86.6)	94 (80.3)	0.339

**Table 2 ijerph-19-06652-t002:** Change in the single health status index scores on the EQ-5D-5L.

Variables	Using Self-Management (*n* = 84)	Not Using Self-Management *(n* = 119)	*p*-Value
Single Health Status Index Score			
Change, mean (SD)	0.08 (0.21)	-0.01 (0.20)	0.002
2021	0.70 (0.21)	0.64 (0.24)	0.089
2022	0.78 (0.16)	0.63 (0.24)	<0.001
Dimension 1: Mobility			
2021	1.90 (1.36)	2.10 (1.37)	0.314
2022	1.51 (0.95)	2.12 (1.35)	0.001
Dimension 2: Self-Care			
2021	1.42 (0.89)	1.63 (1.13)	0.15
2022	1.20 (0.71)	1.66 (1.18)	0.002
Dimension 3: Usual Activities			
2021	1.69 (1.02)	2.08 (1.15)	0.015
2022	1.48 (0.84)	2.12 (1.22)	<0.001
Dimension 4: Pain/Discomfort			
2021	2.15 (0.91)	2.41 (1.07)	0.075
2022	1.93 (0.90)	2.54 (1.10)	<0.001
Dimension 5: Anxiety/Depression			
2021	1.81 (0.83)	1.94 (1.01)	0.327
2022	1.51 (0.70)	2.04 (1.02)	<0.001

## Data Availability

The datasets used and/or analyzed during the current study may be obtained from the corresponding author upon reasonable request.

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
