# Peer review of "Improvement in Quality of Life through Self-Management of Mild Symptoms during the COVID-19 Pandemic: A Prospective Cohort Study"

_ijerph, 2022, doi:10.3390/ijerph19116652_

Round 1
Reviewer 1 Report
In this manuscript, the authors preformed a cohort study to investigate the relationship between the quality of life and the self-management of mild covid symptoms in elder population. The introduction of this research and corresponding results were clearly described. Overall, this manuscript was well prepared, however, it still can be improved with some minor revisions.
The question I have is what is the reason for the elder people to choose using self-management or not using self-management during the pandemic? Is there any questions or variables can be used to indicate this? is there any domestic reason for these people choosing to not use self-management? Or different types of chronic diseases? This should be discussed in the paper.
Author Response
Responses to the reviewers’ comments
Thank you very much for reviewing our manuscript and providing suggestions for improving it. We have provided point-by-point responses to the reviewers’ comments; our revisions are indicated in red font here and in the document. We hope that the revised manuscript meets the journal’s requirements and is now suitable for publication.
Reviewer 1
In this manuscript, the authors preformed a cohort study to investigate the relationship between the quality of life and the self-management of mild covid symptoms in elder population. The introduction of this research and corresponding results were clearly described. Overall, this manuscript was well prepared, however, it still can be improved with some minor revisions.
The question I have is what is the reason for the elder people to choose using self-management or not using self-management during the pandemic? Is there any questions or variables can be used to indicate this? is there any domestic reason for these people choosing to not use self-management? Or different types of chronic diseases? This should be discussed in the paper.
Response:
Thank you for your valuable suggestions. We agree with you regarding adding the description of people who were not motivated to engage in self-management and related factors. We have added these relationships in the Discussion part as follows:
“Information regarding the assessment of and approaches to mild symptoms can be vital for the self-management of them. Health literacy and isolation could have affected the behaviors of those who were not motivated to engage in self-management of mild symptoms in this study. Rural societies have been suffering from isolation due to the COVID-19 pandemic, causing poor health outcomes [43]. Additionally, rurally isolated people with frailty may face difficulties while approaching information resources because of chronic disease(s) [43].Rural social resources regarding knowledge of health management and HSBs should be continuously informed and used to facilitate the efficient self-management of symptoms. Health care professionals should support self-management in these individuals with the collaboration of volunteers in communities who motivate people to manage health with effective exercise and nutrition [44,45].” (Line 204 to 215).
Reviewer 2 Report
Greetings! I had the pleasure of reviewing your manuscript and have a couple of suggestions for you.
First, I believe the article would be strengthened by defining HSB's in the context of the categories in the EQ-5D-5L. Regarding mobility, for example, what is a self-care HSB one might engage in for mild symptoms of mobility? Help the reader get a clear picture of the problems older adults in rural settings face, and what they do to care for themselves. In article (#29 is the intext citation) there was an explanation of how the nurse reviewed health journals to better understand how individuals were engaging in self-care. I believe including a little more description on this type of information would inform the reader because mild symptoms may mean different things to different people.
Second, I also read reference 29 in which it was apparent the two studies are related. I believe it important to state that this study is part of a larger one, if that is indeed the case. Perhaps in the interest of separating the data and the two articles there was less mentioned about the HSBs themselves.
The data analysis looked sound and the discussion was meaningful. I believe the authors need to provide additional explanation of key words or concepts related to "mild symptoms" and "self-care". I imagine they're trying to keep the two articles as distinctly separate as possible, however more information would benefit the understanding of this article.
Author Response
Responses to the reviewers’ comments
Thank you very much for reviewing our manuscript and providing suggestions for improving it. We have provided point-by-point responses to the reviewers’ comments; our revisions are indicated in red font here and in the document. We hope that the revised manuscript meets the journal’s requirements and is now suitable for publication.
Greetings! I had the pleasure of reviewing your manuscript and have a couple of suggestions for you.
First, I believe the article would be strengthened by defining HSB's in the context of the categories in the EQ-5D-5L. Regarding mobility, for example, what is a self-care HSB one might engage in for mild symptoms of mobility? Help the reader get a clear picture of the problems older adults in rural settings face, and what they do to care for themselves. In article (#29 is the intext citation) there was an explanation of how the nurse reviewed health journals to better understand how individuals were engaging in self-care. I believe including a little more description on this type of information would inform the reader because mild symptoms may mean different things to different people.
Response:
Thank you for your valuable suggestions. We agree with your comment regarding defining help-seeking behaviors, self-management and self-care in the context of the EQ-5D-5L. We have added explanations, especially those pertaining to the five dimensions, including self-care, of the EQ-5D-5L as follows:
“The question regarding mobility asks about problems in walking about. Regarding self-care, the question asks about problems in walking or dressing oneself. Regarding usual activities, the question asks about problems in doing one’s usual activities, such as work, study, housework, familial duties or leisure activities. Regarding pain/discomfort, the question asks about having pain or discomfort in one’s usual life. Finally, the question regarding anxiety/depression asks about any feelings of anxiety/depression.” (Line 115 to 120).
Second, I also read reference 29 in which it was apparent the two studies are related. I believe it important to state that this study is part of a larger one, if that is indeed the case. Perhaps in the interest of separating the data and the two articles there was less mentioned about the HSBs themselves.
Response:
Thank you for your valuable suggestions. We agree with the suggestions regarding issue of previous research. Our previous study was conducted to establish the use of this questionnaire for subsequent studies, including this study. We used different databases in the previous study and this one. Therefore, we have elaborated on the previous study in the Method section as follows:
“A validated questionnaire with Spearman’s ρ = 0.707 and Cohen’s kappa = 0.836 was used to measure trends in participants’ HSBs for mild symptoms [29]. In a previous study, we established the use of the questionnaire to inquire about HSB for mild symptoms among older people and checked its validity and reliability. In this questionnaire, participants reported their behavioral responses to mild symptoms. The previous study used data from older people in various settings, such as clinics and hospitals. In this study, we focused on rural communities and defined the presence of self-management in the questionnaire as showing a preference to self-manage mild symptoms.” (Line 129 to 137).
The data analysis looked sound and the discussion was meaningful. I believe the authors need to provide additional explanation of key words or concepts related to "mild symptoms" and "self-care". I imagine they're trying to keep the two articles as distinctly separate as possible, however more information would benefit the understanding of this article.
Response:
Thank you for your valuable suggestions. We agree with them and have revised the suggested parts. We hope that the revised manuscript meets the journal’s requirements and is now suitable for publication.
Reviewer 3 Report
The manuscript presents the results of a study aimed to improve in Quality of Life Through Self-Management of Symptoms During the COVID-19 Pandemic. This topic is interesting because it allows for interventions to empower the population.
Thank you very much for your article but I have identified some issues that are not clear.
I think the title needs to be rewritten because in the abstract the authors indicate that “self-management of mild symptoms”.
It would be necessary to define the concept “self-management of mild symptoms” and “quality of life”
Methods
Although the authors indicate that the Japanese version is validated, they should provide in the text values for the validity and reliability of the Japanese version of EQ-5Q-5L and HSBs. In this way, they do not oblige the reader to read the full article included in the reference.
Thank you very much
Author Response
Responses to the reviewers’ comments
Thank you very much for reviewing our manuscript and providing suggestions for improving it. We have provided point-by-point responses to the reviewers’ comments; our revisions are indicated in red font here and in the document. We hope that the revised manuscript meets the journal’s requirements and is now suitable for publication.
The manuscript presents the results of a study aimed to improve in Quality of Life Through Self-Management of Symptoms During the COVID-19 Pandemic. This topic is interesting because it allows for interventions to empower the population.
Thank you very much for your article but I have identified some issues that are not clear.
I think the title needs to be rewritten because in the abstract the authors indicate that “self-management of mild symptoms”.
Response:
Thank you for your valuable suggestion regarding the title, which. we agree with. We have revised the term used in the title from “symptoms” to “mild symptoms” (Line 3).
It would be necessary to define the concept “self-management of mild symptoms” and “quality of life”
Response:
Thank you for your valuable suggestions. We agree with the suggestion regarding the definition of “QOL”, and have added the definition of “QOL” in the Introduction (Line 48 to 49). Regarding mild symptoms, we have added the definition of “mild symptoms” and “self-mangement” in the introduction and method section. (Line 35 to 36, and 140 to 141)
Methods
Although the authors indicate that the Japanese version is validated, they should provide in the text values for the validity and reliability of the Japanese version of EQ-5Q-5L and HSBs. In this way, they do not oblige the reader to read the full article included in the reference.
Response:
Thank you for your valuable suggestions. We agree with your recommendation regarding the validity and reliability of the questionnaires, which we have added in the Method section as follows:
“These scores were subsequently combined into five-digit numbers representing the participants’ health status profiles. Health status profiles were converted into single health status index scores through the application of a formula that attached values to each response. The Japanese version of the EQ-5D-5L has been validated (R2 = 0.977) [28].
2.3.2. Independent Variable
A validated questionnaire with Spearman’s ρ = 0.707 and Cohen’s kappa = 0.836 was used to measure trends in participants’ HSBs for mild symptoms [29]. In a previous study, we established the use of the questionnaire to inquire about HSB for mild symptoms among older people and checked its validity and reliability. In this questionnaire, participants reported their behavioral responses to mild symptoms. The previous study used data from older people in various settings, such as clinics and hospitals. In this study, we focused on rural communities and defined the presence of self-management in the questionnaire as showing a preference to self-manage mild symptoms.” (Line 126 to 140).
Thank you again for your valuable comments regarding our paper, which we now hope is suitable for manuscript publication.
This manuscript is a resubmission of an earlier submission. The following is a list of the peer review reports and author responses from that submission.
Round 1
Reviewer 1 Report
The manuscript deals with a topic of interest in the COVID-19 pandemic era, namely the difficulties of access to care by the aged inhabitants of rural areas of Japan. The Authors, by a prospective cohort study (demographic data and quolity of life-QOL-collected by a questionnaire in 2021and 2022) showed that "the self-management of the symptoms" is useful in improving QOL among the affected people. The Authors conclude suggesting the need of educational interventions regarding the self management in this kind of population. While dealing with a current problem, the study has important limitations, of which the Authors are aware, i.e the prospective character of the study and the inadequacy of the demographyc data in both the study and control groups.
Reviewer 2 Report
The entire study is based solely on survey analysis. The results do not provide any new and unexpected information or knowledge.